# A Neolithic mega-tsunami event in the eastern Mediterranean: Prehistoric settlement vulnerability along the Carmel coast, Israel

Gilad Shtienberg[1]*, Assaf Yasur-Landau[2,3], Richard D. Norris[4], Michael Lazar[5], Tammy M. Rittenour[6], Anthony Tamberino[7], Omri Gadol[8], Katrina Cantu[4], Ehud Arkin-Shalev[2,3], Steven N. Ward[9], Thomas E. Levy[1,7]

1 Department of Anthropology, Scripps Center for Marine Archaeology, University of California, San Diego, California, United States of America, 2 Department of Maritime Civilizations, L.H. Charney School of Marine Sciences, University of Haifa, Haifa, Israel, 3 The Recanati Institute for Maritime Studies (RIMS), University of Haifa, Haifa, Israel, 4 Scripps Center for Marine Archaeology, Scripps Institution of Oceanography, University of California, San Diego, California, United States of America, 5 Dr. Moses Strauss Department of Marine Geosciences, L.H. Charney School of Marine Sciences, University of Haifa, Haifa, Israel, 6 Department of Geosciences, Utah State University, Logan, Utah, United States of America, 7 Levant and Cyber-Archaeology Laboratory, Scripps Center for Marine Archaeology, University of California, San Diego, California, United States of America, 8 The Hatter department of Marine Technologies, University of Haifa, Haifa, Israel, 9 Institute of Geophysics and Planetary Physics, University California Santa Cruz, Santa Cruz, California, United States of America

* Gshtienberg@ucsd.edu

**Data Availability Statement:** All relevant data are within the manuscript and its Supporting Information files.

## Abstract

Tsunami events in antiquity had a profound influence on coastal societies. Six thousand years of historical records and geological data show that tsunamis are a common phenomenon affecting the eastern Mediterranean coastline. However, the possible impact of older tsunamis on prehistoric societies has not been investigated. Here we report, based on optically stimulated luminescence chronology, the earliest documented Holocene tsunami event, between 9.91 to 9.29 ka (kilo-annum), from the eastern Mediterranean at Dor, Israel. Tsunami debris from the early Neolithic is composed of marine sand embedded within fresh-brackish wetland deposits. Global and local sea-level curves for the period, 9.91–9.29 ka, as well as surface elevation reconstructions, show that the tsunami had a run-up of at least ~16 m and traveled between 3.5 to 1.5 km inland from the palaeo-coastline. Submerged slump scars on the continental slope, 16 km west of Dor, point to the nearby "Dor-complex" as a likely cause. The near absence of Pre-Pottery Neolithic A-B archaeological sites (11.70–9.80 cal. ka) suggest these sites were removed by the tsunami, whereas younger, late Pre-Pottery Neolithic B-C (9.25–8.35 cal. ka) and later Pottery-Neolithic sites (8.25–7.80 cal. ka) indicate resettlement following the event. The large run-up of this event highlights the disruptive impact of tsunamis on past societies along the Levantine coast.

## Introduction

Recent destructive events such as the 2004 Sumatra Tsunami, have increased awareness of the need to better understand the generating mechanisms, recurrence intervals and possible

**Funding:** The Authors gratefully acknowledge the generous support provided by Scripps Center for Marine Archaeology, Scripps Institution of Oceanography, UC San Diego; The Koret Foundation (Grant ID 19-0295); Murray Galinson San Diego – Israel Initiative; the Israel Institute (Washington, D.C.); Marian Scheuer-Sofaer and Abraham Sofaer Foundation; Norma and Reuben Kershaw Family Foundation; Ellen Lehman and Charles Kennel - Alan G Lehman and Jane A Lehman Foundation; Paul and Margaret Meyer and the Israel Science Foundation (Grant ID 495/18).

**Competing interests:** We declare that we the authors have declared that no competing interests exist.

outcomes of catastrophic tsunamis for natural hazard assessments. Similar interest has developed within the geoarchaeological community with renewed attention to the impact of past natural disasters on cultural development and human history. Nevertheless, studies that investigate the interconnected influences of catastrophic processes and their impact on prehistoric settlement regimes and cultural progression are less common [1].

In the eastern Mediterranean (Fig 1), tsunamis are frequent, occurring at a rate of around eight events per century in the Aegean region over the past ca. 2,000 years [2] and approximately ten per century over the past 3,000 years in the Levant basin [3–6]. Most of these events are small and have only local impacts. However, 23 tsunami deposits have been recognized in the region from the past ca. 6,000 years [7], averaging one event large enough to leave tsunami sedimentary facies every ~160 years, suggesting that this is a widespread phenomenon.

While the tsunami frequency since Roman times is well known, the record of earlier events is less well defined, with only five documented Holocene occurrences before 2,500 years and no events identified prior to 6,000 years ago in the Levant basin (Fig 1; S1 and S2 Tables). Palaeo-tsunami coastal deposits [7,10] have been shown to be a useful environmental indicator when integrated with historical records and can shed light on the landward extent of tsunami transgressions, run-up height [11], timing of events, and possible identification and evaluation of triggering mechanisms [12]. Tsunami deposits and storm deposits can be differentiated by their sedimentological characteristics such as grain size, facies homogeneity, sediment grading morphology, internal micro—macro faunal remains as well as inland depositional extent and relation to the local sea level during deposition [13]. Here, we describe a large early Holocene tsunami deposit in coastal sediments from Dor in NW Israel (Fig 2), as well as its probable prehistoric socio-cultural impact.

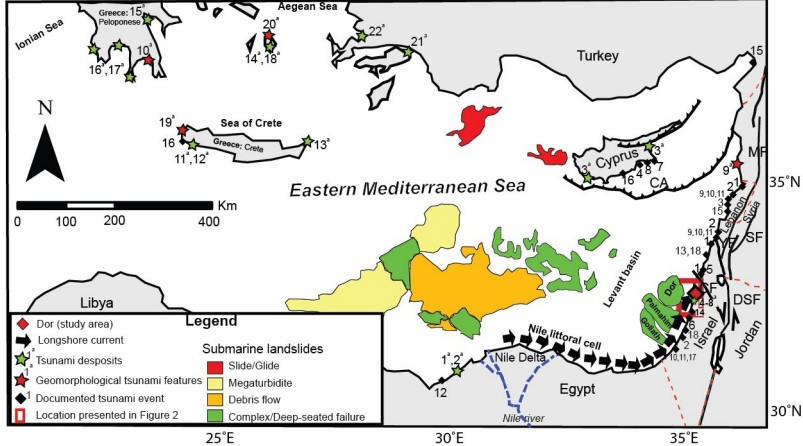

**Fig 1. Geological sketch of the eastern Mediterranean modified after natural earth (https://www.naturalearthdata.com in the public domain), showing the main near-shore sediment transport mechanism (black arrows), selected thrusts (CA–Cypriot Arc) and major fault lines (CF- Carmel fault; DSF- Dead Sea Fault system; SF- Seraghaya fault; MF-Missyaf fault; YF-Yammaounch fault; [3,8], submarine landslides as well as tsunami deposits, geomorphological tsunami features and documented tsunami events (Modified from [9]).** The name compilation of the sites presented in the figure are: 1[a]-2[a] (Alexandria); 3[a] (Paphos, Polis, Cape, Greco); 4[a]-8[a] (Caesarea Marittima, Jiser al-Zarka); 9[a] (Byblos, Senani Island); 10[a] (Elos); 11[a] (Gramvousa, Balos, Falasarna, Mavros, Stomiou, Gramenos, Paleochora); 12[a] (Western Crete); 13[a] (Palaikastro); 14[a] (Pounta); 15[a] (Limni Moustou); 16[a] (Pylos, Porto Gatea, Archangelos, Elaphonisos); 17[a] (Limni Divariou); 18[a] (Santorini); 19[a] (Balos bay); 20[a] (Thera); 21[a] (Dalaman); 22[a] (Didim) for the previously dated tsunami deposits and 1 (Lebanon, Israel, Syria); 2 (levant coast); 3 (Paphos, Polis, Cape, Greco); 4 (S-E Cyprus); 5 (Akko); 6 (Yaffo); 7–8 (S-E Cyprus); 9–11 (Levant coast); 12 (The Nile cone); 13 (Lebanon); 14 (Levant coast); 15 (southern turkey); 16 (Cyprus); 17 (Israel); 18 (Lebanon–Israel) for the previously dated tsunami events. Further details regarding the tsunami data are discussed in S1 and S2 Tables.

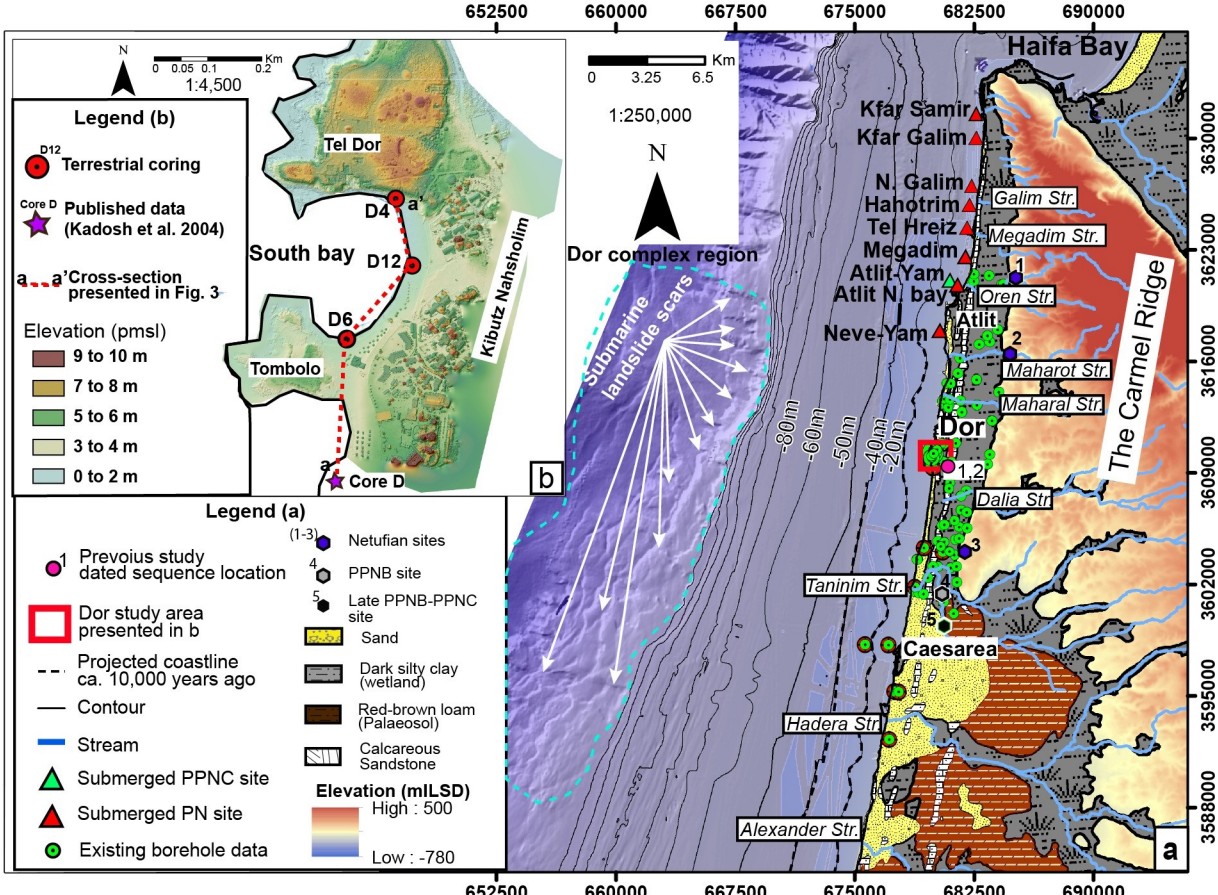

**Fig 2. Location maps.** (a) Israel's Carmel coastal plain: surface lithologies, streams, shelf bathymetry and elevations (Republished from [14] under a CC BY license, with permission from [the geological Survey of Israel], original copyright [1994]). The red and green triangles indicate location of Neolithic habitations based on Galili et al. [15] while the red square annotates the location of the study area. The numbered red circles represent previously studied zones in which the stratigraphic sequence was investigated and is described in the following papers according to their numbering: (1) Kadosh et al. [16]; (2) Sivan et al. [16]. The numbered hexagons annotate prehistoric sites (1) Nahal Oren (Natufian period); (2) El-wad (Natufian period); (3) Kebara (Natufian period); (4) Tel Mevorakh (PPNB period); (5) Aviel (late PPNB-PPNC period). The projected coastline during the tsunami at ca 9,910–9,290 ya ca. 9.91–9.29 ka, is presumed to have been located between at about 40 to 16 m below present-day sea-level and 3.5–1.5 km west of the current shoreline. (b) The coast of Dor with existing cores and new drilling locations as well as elevations.

Tel Dor, located along the Carmel coast of NW Israel, is a maritime city-mound that has been occupied from the Middle Bronze II period (ca. 2000 to 1550 BCE) throughout the Roman period (third century AD) while Byzantine and Crusader remains are also found on the tel. The local environment of Dor is characterized by a series of unique embayments/pocket beaches that stand out from the linear morphology of the southeastern Mediterranean littoral shore face (Figs 1 and 2). The pocket beaches protect this part of the coast from wave energy and help preserve the unique late Pleistocene to Holocene geologic record. The preservation of quaternary deposits and landforms at Dor coupled with the abundance archaeological sites provide a unique opportunity to investigate the geomorphological processes and human-landscape interactions in the eastern Mediterranean.

## Materials and methods

Multidisciplinary analyses of a large spatial and temporal dataset distributed across the Carmel coast was made with detailed site-specific coastal mapping, supplemented with three terrestrial

boreholes and an underwater archaeological excavation conducted in the south Bay of Dor after attaining the proper permits from the Israel antiquities authority and Israel Natural Parks Authority.

## Compilation of existing datasets

Sixty terrestrial and offshore borehole datasets were collected from previously published research and their X, Y and Z coordinates incorporated into a series of GIS (ESRI ArcGIS 10.7) tables. Low resolution terrestrial Digital Elevation Models (DEM) and bathymetric raster files ($25 \times 25$ m bin size), soil maps, rectified historical aerial photographs and chronostratigraphic data were also uploaded into ArcGIS and stored as one homogenous database after modification (Fig 2B).

## Geomorphological mapping and surface model creation of the study area

The coastal geomorphological characteristics and surface lithologies of the study area were mapped in detail through land (2017–2018) and underwater surveys (February 2019). Photogrammetric remote sensing techniques were used to create a digital surface model (DSM) of the study area's terrestrial part. The survey imagery was collected in RAW format and acquired by a heavy-lift octocopter drone carrying a gimbal stabilized Sony A7R digital camera with a 36.4MP full frame sensor, and 28mm focal length lens. The survey data was georeferenced with a South Galaxy G1 RTK-GPS collecting 63 ground control points (GCPs) distributed across the study area with a vertical and horizontal error no greater than ±7 cm. A total of 808 images and 43 GCPs were used and for photogrammetric reconstruction in Agisoft Photoscan on high settings, resulting in a $4.41 \times 4.41$ cm ground sampling distance (GSD) DSM, and an orthorectified photomosaic with an average $2 \times 2$ cm GSD.

## New borehole drilling and petro-sedimentological analysis

The locations of the boreholes cored in the current study were chosen based on understandings gained from previous studies that surveyed the sub-surface lithology of Dor [16,17]. The new locations were sampled over two coring expeditions (24.8.2018 and 3.2.2019) along the shoreline from the northern to southern end of Dor's South Bay (Fig 2B). Terrestrial coring was completed with a Geo-probe 6620DT direct push corer penetrating up to 9 m below the surface until reaching the Pleistocene calcareous sandstone surface. Locations and surface elevations of the boreholes were measured using a Proflex 500 RTK-GPS with precision of ± 1 cm and ± 5 cm, respectively (Fig 3A). The cores were then split lengthwise for color (Munsell color chart) and lithological description. Digital photographic analysis of the sediments assigned brightness values ranging from 0 to 100 for each pixel. Elemental variation along the 3 cores was done in 1 cm resolution in an Avaatech X-ray fluorescence (XRF) analysis, in an excitation voltage of 10 and 35 kV, with a 2 cm diameter beam. The raw element values (photon counts) were then normalized to silica values, a dominant element in Israel's coastal sediments, to enable relative difference assessment for each facies/unit sample. Particle-size distribution (PSD) and magnetic susceptibility (MS) measurements were carried out on 39 samples from core D4 spanning across the identified lithological units, which were identified in the newly obtained cores. Twelve additional PSD measurements were conducted on core D6 strengthening the stratigraphical correlation. The PSD analysis were conducted with Malvern Instruments Mastersizer 2000 laser particle size analyzer while the MS analysis was done using a Barington Magnetic Susceptibility MS3 System. Faunal analysis was carried out on 14 selected samples spanning across the identified lithological units from Borehole D4. Samples

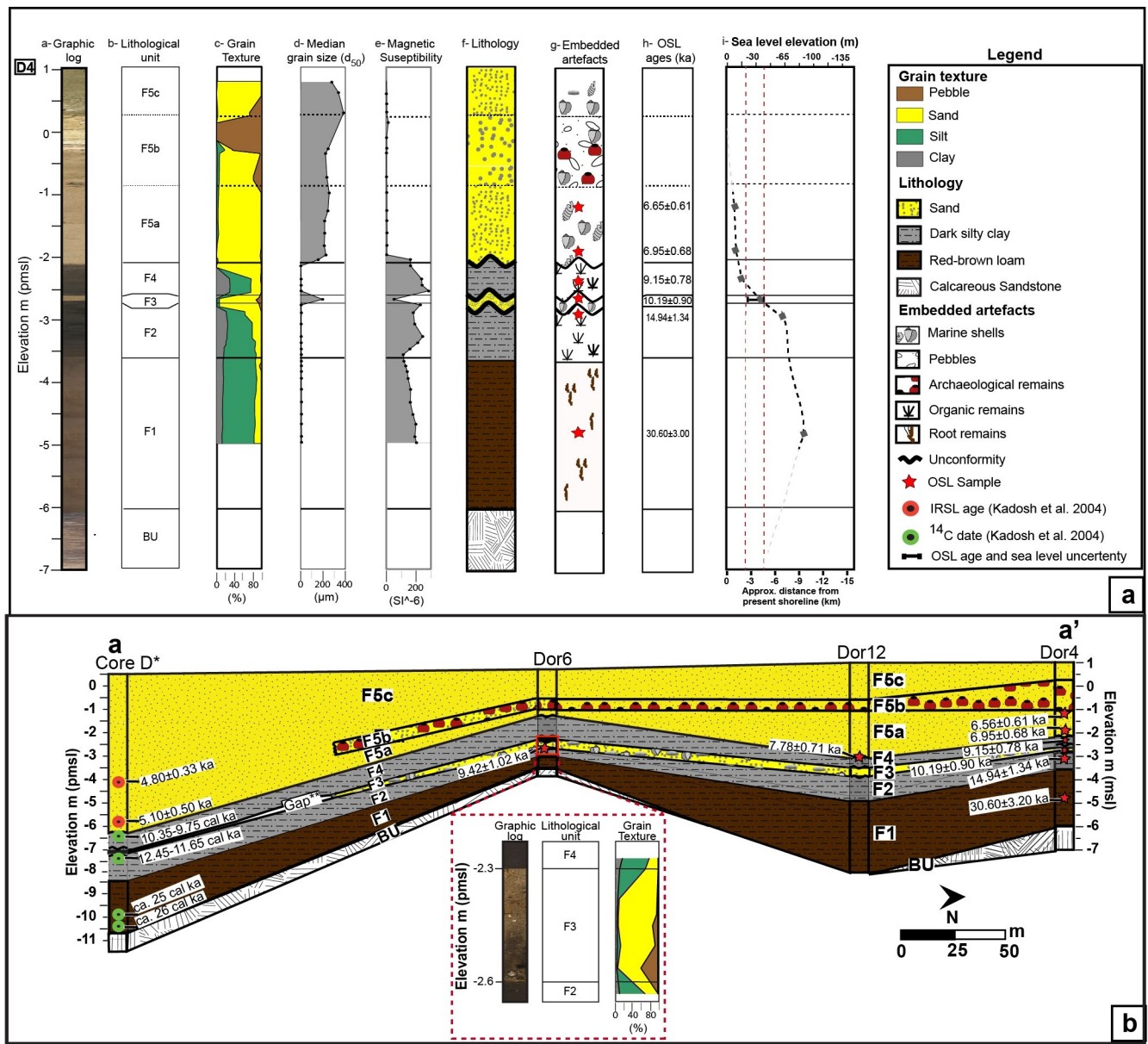

**Fig 3. Core analysis and chronostratigraphic correlation.** (a) Borehole D4 with core scan; lithological unit name; sedimentological and petrophysical results; lithological identification with OSL sampling location; OSL ages presented before 2018; and corresponding sea level [23,24] as well as approximate shoreline location. (b) Chronostratigraphic cross sections in the coastal area of Dor based on sedimentological and OSL data obtained in the present study, presented for thousand years ago (ka, marked with a red star) as well as shtienberg et al. (submitted; blue star) correlated with the lithological results, and [14]C calibrated dates (cal. ka; green circles) published in Kadosh et al. [16]. A closeup of units F2, F3 and F4 in core D6 with its lithological unit name and accumulative grain texture results. The modern topography portrayed in the cross sections was extracted from the DEM presented in Fig 2B. See Fig 2B for cross section location.

of ~10 g dry weight were washed over a 500/150/63μm sieves. Paleontological distribution was then identified with a microscope.

## Optically stimulated luminescence (OSL) dating

OSL dating was determined to be the most appropriate dating method for the cores because of the quartz-sand rich nature of the sediments and known problems with radiocarbon dating at

the site [18]. Over all eight samples for OSL dating were collected from Boreholes D4, D6 and D12 and were processed under dim orange light (~590 nm) at the Utah State University Luminescence Laboratory. The 125–212 μm quartz sand fraction was purified using 10% HCl and H2O2 to remove carbonates and organics, sodium polytungstate (2.72 g/cm3) to remove heavy minerals and concentrated HF and HCl acids to remove feldspars, etch the quartz grains and prevent formation of fluorite precipitates.

Samples for dose rate determination were collected from above, below and within the targeted depths within the core. Conversion factors of Guérin et al. [19] were used to calculate dose rates based on concentrations of ICP-MS and ICP-AES analyzed K, Rb, Th and U. Cosmic contribution was calculated using sample depth, elevation, latitude/longitude and water content [20].

OSL analysis followed the single-aliquot regenerative (SAR) dose method on small aliquots (1 mm) of quartz sand [21]. Equivalent doses and ages were calculated using the central age model [22]. OSL ages and dose rate information are presented in Tables 1 and 2 (see supplementary information for more details).

## Underwater excavation

To supplement the paleo-environmental data retrieved from coring around and in the South Bay of Dor, under water excavations were carried out (February 2019) to reveal archaeological features related to human settlement history in the area. A water dredge system was used powered by an Angus Fire diesel pump installed on a small fishing boat at a maximum depth of 2 meters. Excavation was conducted by team of divers. Artifacts were located and collected underwater, then recorded and registered on the coast by a designated registrar. Architecture was recorded underwater using both by hand and 3D photogrammetry. Accurate positioning of the finds was conducted with the use of a Leica TS02 total station in tandem with a RTK–GPS South Galaxy G1 operated from land, with the bottom of the reflector pole being positioned directly over the underwater remains.

## Results

The cores penetrated as deep as 9 meters below the surface reaching tan to orange brittle calcareous sandstone (Fig 3) and interpreted as aeolian sandstone [23]. The aeolianite is overlain by red-brown loam with OSL age of 31.59 ± 3.20 ka (Unit F1; Fig 3; S1 and S2 Tables); the loam is barren of any flora and fauna and contains irregularly shaped hard calcareous and manganese nodules consistent with palaeosols that are widespread in the area [9]. Late Pleistocene to early Holocene-age sediments OSL dated to 14.94 ± 1.34 ka and 9.42 ± 0.85 ka, from samples received from the middle part of unit and its surface respectively, include dark greybrown homogenous organic-rich loam (Unit F2; Fig 3; Tables 1 and 2) with brackish microfauna and fresh to brackish plant remains (S1 Dataset) interpreted as fresh—brackish wetland deposits. The upper part of the wetland unit is truncated by an abrupt contact and is superimposed by a 20–35 cm thick poorly sorted light-yellow quartz sand layer (Unit F3; Fig 3). The sand deposit has an OSL age of 10.19 ± 0.90 ka (Tables 1 and 2) and contains marine mollusk shells, including *Glycymeris* and *Tucetona* bivalves and *Conomurex persicus* gastropods. The marine faunal remains are complemented with limestone pebbles, calcareous sandstone clasts (Fig 4) and dark silty clay rip-up clasts identical in color and texture to the underlying fresh to brackish wetland deposits (Unit F2). The characteristics of Unit F3 suggests that this unique sand layer was deposited during and abrupt event such as a storm or tsunami. We discuss the possible origin of this sand interbed in more detail below (and extended results section in supplementary information). A grey-brown homogenous organic-rich silty loam overlies the

**Table 1. Optically stimulated luminescence (OSL) age information.** OSL ages are presented as thousands of years (ka) before 2018.

| Sample num. | USU num. | Num. of analyses[1] | Dose rate (mGy/yr) | Equivalent Dose[2] ± 2σ (Gy) | OSL age ± 1σ (ka) |
|---|---|---|---|---|---|
| **D4L-B 55-65cm** (1.16–1.25 cm core length) | USU-2952 | 18 (20) | 0.64 ± 0.03 | 4.17 ± 0.25 | **6.56 ± 0.61** |
| **D4L-C 45-55cm** (1.85–1.95 cm core length) | USU-2953 | 17 (19) | 0.63 ± 0.03 | 4.38 ± 0.38 | **6.95 ± 0.68** |
| **D4L-C 85-100cm** (2.25–2.4 cm core length) | USU-2954 | 16 (21) | 1.64 ± 0.06 | 15.01 ± 0.80 | **9.15 ± 0.78** |
| **D4L-D 2.5-11cm** (254.5–263 cm core length) | USU-2955 | 26 (36) | 0.53 ± 0.02 | 5.42 ± 0.31 | **10.19 ± 0.90** |
| **D4L-D 105-120cm** (357–372 cm core length) | USU-2956 | 17 (21) | 1.50 ± 0.06 | 22.44 ± 1.76 | **14.94 ± 1.34** |
| **D4L-F 90-105cm** (466–471 cm core length) | USU-2957 | 19 (26) | 1.58 ± 0.06 | 48.39 ± 6.50 | **30.59 ± 3.20** |
| **DL12C 75-85cm** (392–402 cm core length) | USU-3300 | 22(34) | 0.64 ± 0.03 | 4.98 ± 0.31 | **7.78 ± 0.71** |
| **DL6C 20-30cm** (325–335 cm core length) | USU-3305 | 17(20) | 1.44 ± 0.06 | 13.59 ± 1.09 | **9.42 ± 0.85** |

[1] Age analysis using the single-aliquot regenerative-dose procedure of Murray and Wintle [21] on 1-mm small-aliquots of quartz sand unless otherwise noted. Number of aliquots used in age calculation and number of aliquots analyzed in parentheses. Datum for ages is AD 2018.

[2] Equivalent dose ($D_E$) calculated using the Central Age Model (CAM) unless otherwise noted.

sharp upper contact of the sand bed and is interpreted as the continuation of wetland deposits (Unit F4; Fig 3). Unit F4 is OSL dated to 9.15 ± 0.78 ka and 7.78 ± 0.71 ka (Tables 1 and 2) from samples received from the bottom half part of unit and its surface respectively (Fig 3B). Finally, the whole sequence is capped with 2–4 m of sandy marine deposits (Unit F5) that commenced deposition in 6.95 ± 0.78 ka and constitute the deposits of the modern beach in Dor (Fig 3; Tables 1 and 2).

**Table 2. Dose rate information for OSL samples selected from borehole D4, D12 and D6.**

| Sample num. | USU num. | Depth (m) | In-situ H₂O (%) | K (%)[1] | Rb (ppm)[1] | Th (ppm)[1] | U (ppm)[1] | Cosmic (Gy/ka) |
|---|---|---|---|---|---|---|---|---|
| **D4L-B 55-65cm** (1.16–1.25 cm core length) | USU-2952 | 2.15–2.32 | 18.1% | 0.35±0.01 | 7.8±0.3 | 0.5±0.2 | 0.9±0.1 | 0.14±0.01 |
| **D4L-C 45-55cm** (1.85–1.95 cm core length) | USU-2953 | 2.85–2.95 | 13.0%[2] | 0.49±0.01 | 10.9±0.4 | 0.5±0.2 | 0.4±0.1 | 0.13±0.01 |
| **D4L-C 85-100cm** (2.25–2.4 cm core length) | USU-2954 | 3.25–3.4 | 32.9% | 1.43±0.04 | 50.0±2.0 | 6.0±0.5 | 0.9±0.1 | 0.12±0.01 |
| **D4L-D 2.5-11cm** (254.5–263 cm core length) | USU-2955a[3] USU-2955b | 3.625–3.71 | 91.1% | 0.56±0.01 1.05 ±0.03 | 15.5±0.6 37.1 ±1.5 | 1.2±0.1 4.4 ±0.4 | 0.6±0.1 0.9 ±0.1 | 0.12±0.01 |
| **D4L-D 105-120cm** (357–372 cm core length) | USU-2956 | 3.705–3.815 | 32.7% | 1.25±0.03 | 49.6±2.0 | 6.3±0.6 | 1.0±0.1 | 0.12±0.01 |
| **D4L-F 90-105cm** (466–471 cm core length) | USU-2957 | 5.7–5.85 | 30.1% | 1.34±0.03 | 48.6±1.9 | 6.0±0.5 | 1.1±0.1 | 0.10±0.01 |
| **DL12C 75-85cm** (392–402 cm core length) | USU-3300[4] | 3.75–3.85 | 19.1%[5] | 0.65±0.02 0.29 ±0.01 | 20.6±0.8 7.3 ±0.3 | 1.7±0.3 0.6 ±0.1 | 0.6±0.1 0.8 ±0.1 | 0.12±0.01 |
| **DL6C 20-30cm** (325–335 cm length) | USU-3305 | 3.2–3.3 | 24.5%[5] | 1.15±0.03 | 1.15±0.03 | 5.1±0.5 | 1.2±0.1 | 0.13±0.01 |

[1] Radioelemental concentrations determined using ICP-MS and ICP-AES techniques; dose rate is derived from concentrations by conversion factors from Guérin et al. [19]. Grain size for all samples is 125–212μm.

[2] Moisture content affect by drying of the core. 20% moisture content assumed over burial history and used in dose rate calculation.

[3] Beta DR uses 100% of chemistry from USU-2955a (flood layer), gamma DR uses 75% from USU-2955a and 25% from USU-2955b (underlying silt).

[4] Radioelemental concentrations for samples were averaged in dose rate calculation: above sample (upper values) and below sample (lower values).

[5] A value of 30 wt% water content use for these samples. This value was obtained from a companion core that was sub-sampled previously. The values here are assumed to be underestimates due to drying over time.

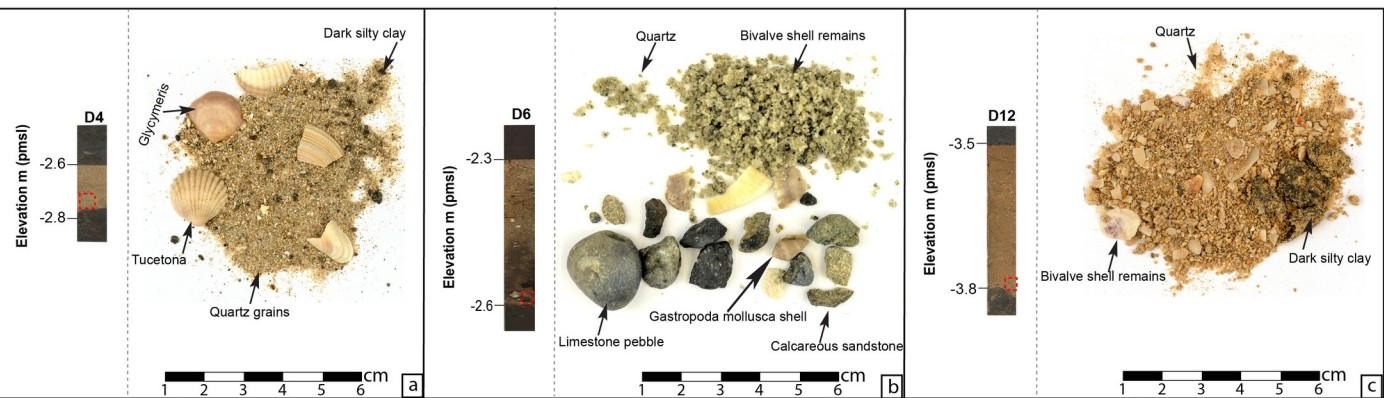

**Fig 4. The sedimentological features of Unit F3 presented for Boreholes D4 (panel-a), D12 (panel-b), D6 (panel-c) with their relevant core extraction depth (red dashed square).** The location of the boreholes is displayed in Fig 2B.

## Discussion

The age of the abrupt marine sand interbed (Unit F3) including its uncertainties (10.19 ± 0.90 ka; Table 1) and age constraint from the underlying wetland surface (unit F2; 9.42 ± 0.85 ka; Fig 3B; Table 1) as well as overlying wetland bottom (unit F4; 9.15 ± 0.78 ka) indicates that deposition occurred between 9.91 to 9.29 ka (see Fig 5 for further details) when global sea-level was ca. 40–16 m below present sea level (Fig 3A; ref. [24–26]). The early Holocene shoreline at

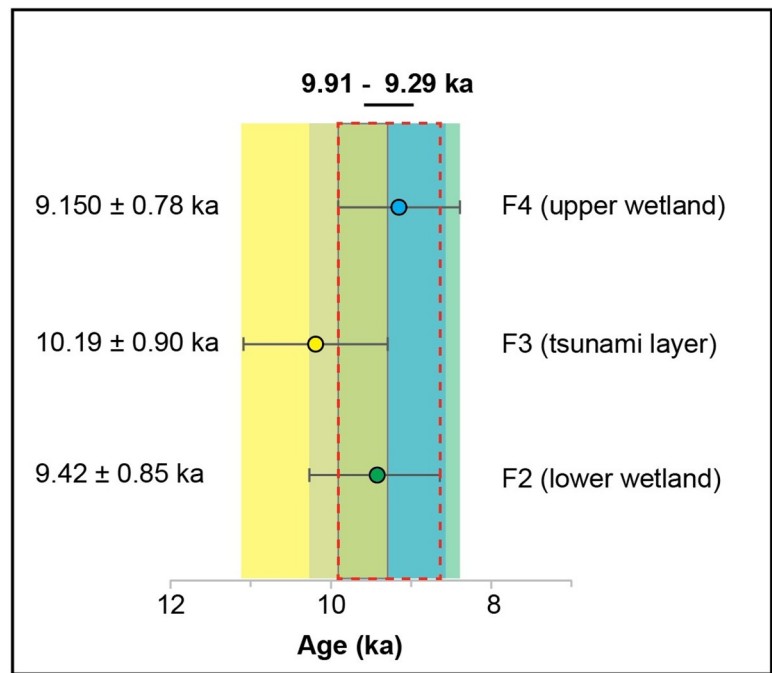

**Fig 5. Age constraint for the tsunami deposit (Unit F3) based on the ages and stratigraphic position of the lower wetland deposit (Unit F2) the abruptly overlying sandy tsunami deposit (Unit F3) and upper wetland deposit (Unit F4) that have been correlated between cores in the study area (Fig 3).** The age constraint for the tsunami deposit (Unit F3) is based on the overlapping ages and uncertainties (highlighted by dashed red line) between the Unit F2 wetland deposit (green circle, core D6) the overlying Unit F3 tsunami deposit (yellow circle, core D4) and superimposing Unit F4 wetland deposit (blue circle, core D4).

this time was plotted against the offshore bathymetric chart after consideration of the Holocene shelf sediment thickness [26] and indicates its location was ~ 3.5–1.5 km seaward from the current shoreline (Figs 2A and 3A). In order to deposit marine shells into the contemporaneous fresh to brackish wetland at Dor, the wave front must have traveled a minimum distance of 1.5 km with a coastal run-up of at least ~16 m. The run-up could have been much larger as the oldest permissible dates on the deposit imply a travel distance of 3.5 km and a run-up of as much as 40 m. The possibility of an extreme winter storm can be ruled out because even the strongest storms only produce surge up to a few hundred meters inland and yield a run-up of only tens of centimeters to a few meters at most [13].

In addition to the estimated run-up, the sand layer (Unit F3) is composed of a poorly sorted sand with marine shells and the rip-up clasts of the underlying wetland deposits, all of which are indicative of a tsunami. Previously identified tsunamis in the eastern Mediterranean from the past ca. 6,000 years (Fig 1, S1 and S2 Tables and ref. [7,10,27]) have had smaller run-up distances as the palaeo event reported here, consisting of inland dispersion limited to only 300 meters compared to the present shoreline. Thus, the Dor event, was generated by a much stronger mechanism than previously documented events in the Eastern Mediterranean.

Local tsunami-generating mechanisms in the Levant basin include earthquakes associated with onshore and offshore faulting as well as submarine landslides, linked to over steepened slopes or earthquake-induced failure. The main source of earthquakes in Israel is the Dead Sea Fault system (DSF; Fig 1) that crosses the Middle East from south to north parallel to the present Levant coast [28]. Although situated onshore, this large seismogenic system is located close to the coastal zone and can generate strong ground motions that could influence failures along the continental margin. The Carmel fault (CF; Fig 1) branches off the Dead Sea Fault in a NW–SE direction extending onto the continental shelf of the Mediterranean Sea. This fault line is believed to be capable of producing earthquakes up to a magnitude (M) of 6.5 [29]. Salamon and Manna [30] empirically constrain the relationship between the magnitude of inland and offshore earthquakes that generate tsunamigenic submarine landslides and the maximal distance of the tsunami source from the epicenter of the seismogenic fault. They estimate the threshold magnitude for tsunamigenic earthquakes to be approximately M 6. Notably, an earthquake contemporary (ca. 10 ka) with the Dor paleo-tsunami has been dated using damaged speleothems from a cave in the nearby Carmel ridge [31]. Given age uncertainty of the event, this earthquake could have triggered an underwater landslide that produced the tsunami recorded here.

Underwater landslides occur when a gravity induced driving stress exceeds a resisting shear strength, this can result in slope failure due to high pore pressure and upslope sedimentary over steepening. Along the continental slope of Israel, a series of landslide-scar complexes are evident between 130 and 1000 m below present sea level, where slopes exceed a critical gradient of 4˚– 5˚ (e.g. Dor and Goliath submarine landslides, Figs 1 and 2A). Similar to the Atlantic Ocean margin [32], a compilation of submarine landslide occurrences around the Mediterranean basin [33] suggest that the occurrence of submarine landslide events was higher during the late Pleistocene to mid-Holocene, and decreased ca. 5 ka. Lee [32] attributed the higher submarine landslide occurrence to eustatic sea level changes and slope instability. In the early stages of post-glacial sea-level rise, the rate of rise is outpaced by the rate of sediment accumulation [33], creating aggregational clinoform patterns and a steepened shelf edge. These over steepened continental slopes can collapse during subsequent sea-level fluctuations [34,35].

Aside from the numerous (> 200) local smaller landslides that dot the Israeli continental shelf, two very large slide complexes have been recognized. These are the Dor and Palmachim submarine complexes located offshore Dor and in the southern parts of Israel's slope (Figs 1 and 2A; e.g. ref. [36,37]). The Dor complex lies in the shallowest part of the shelf break at a

depth of 180–200 m below present sea level (Figs 1 and 2; ref. [36]), ~16 km directly offshore the study area. The Dor complex is clearly visible in multibeam bathymetry (Fig 2A; e.g. ref. [37]) over a 120 km$^2$ area of the continental slope where faults penetrate the entire post Pliocene-Pleistocene sediment sequence [38]. Individually, these complexes are ~10 km wide and seem to interact with mostly offshore dipping faults close to the sea floor suggesting that they are still active. Katz et al. [35] proposed that slide scars found in the Israeli continental slope are younger than 17 ka based on constrained sedimentation rates. A similar mega-structure was identified by Garziglia et al. [39] offshore the western Nile margin extending over an area of 500 km$^2$ with a volume of 14 km$^3$ (Fig 1) and dated to ca. 10 ka, another indication of the prevalence of these features during this time period.

The distance of the Dor landslide to the study area on the Carmel coast, as well as its water depth, could explain the estimated size of the tsunami wave and run-up. Kaldaron-Asael et al. [40] suggest a universal relationship between the area and volume of a landslide to the run-up height of associated tsunamis. Model simulations off the Canary Islands by Abadie et al. [41] suggest that slump volumes of 20, 40 and 80 km3 would produce tsunami heights of 75, 150 and 200 m respectively, 10–20 km from the failure site. These initial waves rapidly decay with distance from the slump source. The Abadie et al. [42] model suggests that slump volumes of a few km$^3$ would be enough to produce the Dor tsunami deposit given a run-up of at least 16 m, 16 km from the Dor complex.

Preliminary computer simulations of the Dor landslide and tsunami using the Tsunami Squares Method [42] produced run-ups of 12–18 m (supplementary material). Using these model simulations, as well as indications of substantially higher occurrences of underwater landslides during the early parts of the Holocene, we suggest that the paleo-tsunami deposit at Dor (Unit F3) is plausibly connected to slope failure in the Dor complex and was perhaps triggered by movements on the Carmel Fault system.

## Impact on neolithic settlements

The tsunami event identified in Unit F3 (whose date range is 9.91–9.29 ka), indicates that it occurred during the middle PPNB (Pre-Pottery Neolithic B) cultural periods, but before the growth of large late PPNB village sites ca. 9.25–8.70 cal. ka [43]. While Early and Middle Epipalaeolithic (22–14 cal. ka) sites are known from the coastal plain in NW Israel, later Natufian (12.5–12 cal. ka) and PPNA (11.7–10.5 cal. ka) sites are almost absent from the same area [44]. The tsunami event, which occurred during the middle PPNB, may account for the absence of late Natufian and PPNA sites in the low-lying coastal areas.

The concentration of Natufian sites above the coastal plain is highlighted by the string of sites along the Carmel mountain, such as at Nahal Oren, El-Wad and Kebara (purple hexagon marked with 1–3 in Fig 2A; ref. [45]). These elevated sites would not have been directly affected by the tsunami because they are at least ~60 m above present sea level [46]. However, a tsunami event with a run-up of at least 16 m is fully capable of dramatically reducing the carrying capacity of the lowland region of the Carmel coast by destroying crop lands, pastures, herds, and near-shore marine fisheries and resources (Fig 2A). With carrying capacity and fertility decreased, population mobility is thought to increase, influencing the Neolithic demographic transition and possibly decreasing population density [47]. The rapid growth in size and area of autonomous Middle PPNB settlements in the hill country of Israel and across Jordan may reflect the move away from the coastal lowlands [48,49].

Based on previous surveys and excavations in the Carmel coastal plain and adjacent submerged environment, there is a paucity of sites from the Natufian to early PPNB periods. A likely explanation for this observation is that the tsunami recorded at Dor removed evidence

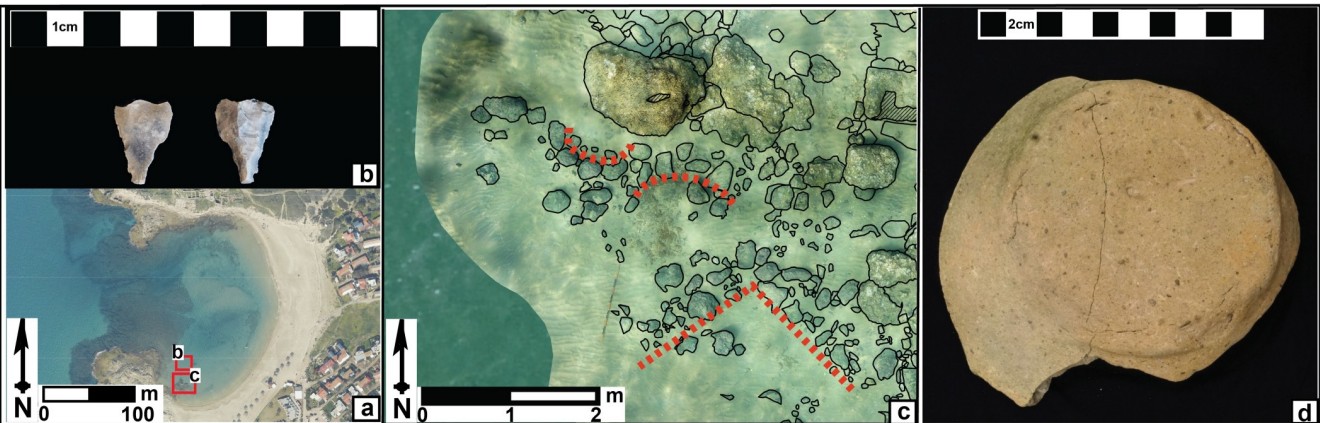

**Fig 6.** Neolithic remains and construction uncovered underwater in the south bay of Dor during joint excavations conducted by the Department of Maritime Civilizations at the University of Haifa and Scripps Center of Marine Archeology at the University of California San Diego: (a) location of the findings marked by red polygons—each letter is attributed to the finds presented in the next parts of the figure; (b) Pre pottery Neolithic–Early Pottery Neolithic arrowhead (c) Neolithic architecture which includes curved installations as well as wall foundations; (d) Pottery Neolithic ceramic base excavated in 2019 by the Department of Maritime Civilizations, University of Haifa.

of coastal Natufian through the early PPNB sites. The Dor paleo-tsunami event led to erosion of wetland sediment in the coastal area, where the age contrast between the wetland sediment and the tsunami suggests the loss of 4,000 years of the archaeological record. The absence of both Natufian and PPNA deposits from most of the coastline is consistent with erosion of the coastal zone by the tsunami.

Recovery from the early Holocene tsunami event was slow as evidence for PPNB settlement on the Carmel coast is rare. One exception may be a heavily disturbed PPNB assemblage found 8 km south of Dor at coastal Tel Mevorakh, (grey hexagon marked with 4 in Fig 2A) at an elevation of ca. 12 m above present mean sea level [50]. Coastal settlements are more widely documented during the post-tsunami Late PPNB and the PPNC 9.25–8.35 cal. ka when the shoreline rose to the ~ 16 to 13.5 m below present sea-level contour based on the relative sea level curve (Fig 2A; ref. [24]). Notably, the re-appearance of abundant archaeological sites coincides with the resumption of wetland deposition in the Dor cores. Settlements include the vast (~ 50 hectare) Late PPNB and PPNC settlement at Aviel in the foothills of the Carmel (black hexagon marked with 5 in Fig 2A; ref. [51]) as well as the Late PPNB and PPNC submerged settlement at Atlit Yam (Fig 2A; ref. [14]) found at ~ 8 to 12 m below present sea-level. Coastal settlement continued to the end of the Pottery Neolithic (PN; ca. 8.4–6.5 cal. ka). To date, a cluster of five submerged PN sites have been investigated from Atlit in the north to Neve Yam in the south, while the flint assemblage from the pottery Neolithic period was found at Tel Mevorakh to the east [50]. Our research demonstrates that this cluster continued further south along the Carmel, with a new underwater (1.5 m below present sea level; Fig 6) PN locality at the South Bay of Dor, 12 km to the south of Atlit.

The strong continuity in settlement patterns during the rest of the Neolithic period indicate that major tsunami events did not significantly affect the settlements of the Carmel coast during the period following the Dor tsunami (9.91–9.29 ka) up to the end of the Pottery Neolithic ca. 6.5 cal. ka (e.g. ref. [14,25]). This allowed an unhindered process of Neolithic growth that led to the rise of increasing social complexes in the southern Levant in the following Chalcolithic and Early Bronze Age periods.

## Conclusions

The deep time investigation conducted in the south bay of Dor, Israel has enabled to identify the earliest known Palaeo-tsunami coastal deposit for the Levantine coast. The large early Holocene tsunami occurred between 9.91 to 9.29 ka with an extreme ~16 m wave height and 1.5–3.5 km run-up on the coast of NW Israel. Early Pre-Pottery Neolithic sites were destroyed along the Carmel coast creating a ~4,000-year settlement gap in the Neolithic archaeological record for the area. The historic record of tsunamis—with events taking place about every 160 years—in the eastern Mediterranean, suggests that tsunamis played a destabilizing role in the development of formative coastal maritime societies.

## Supporting information

**S1 Fig. Borehole D4 (location is displayed in Fig 2B) with lithological classification, description, accompanying features, brightness differences, relative elemental concentration variations and OSL data obtained in the present study.**
(DOCX)

**S2 Fig. Borehole D6 (location is displayed in Fig 2B) with lithological classification, description, accompanying features, brightness differences, relative elemental concentration variations and OSL data obtained in the present study.**
(DOCX)

**S3 Fig. Borehole D12 (location is displayed in Fig 2B) with lithological classification, description, accompanying features, brightness differences, relative elemental concentration variations and OSL data obtained in the present study.**
(DOCX)

**S4 Fig. Equivalent dose distributions for OSL samples.** CAM = Central Age Model. OD = Over-dispersion.
(DOCX)

**S1 Table. Compilation of previously dated tsunami deposits occurring along the eastern Mediterranean coast.** The location of these events and deposits is annotated in Fig 1. Ages with * are ones that are presented for (2σ).
(DOCX)

**S2 Table. Compilation of previously dated tsunami events occurring in the eastern Mediterranean.** The location of these events and deposits is annotated in Fig 1.
(DOCX)

**S3 Table. Correlative archaeological period based on Kuijt and Goring-Morris [22].**
(DOCX)

**S1 Dataset. Faunal assemblage identified in borehole D4.**
(XLSX)

**S1 File. Extended luminescence methods and results section.**
(DOCX)

**S2 File. Extended result section.**
(DOCX)

**S3 File. Link to simulated model of the tsunami event.**
(DOCX)

**S4 File. References.**
(DOCX)

## Acknowledgments

The Authors gratefully acknowledge Issac Oglobin from the Department of maritime Civilization of the University of Haifa as well as Amir Yurman and Moshe Bachar, of the University of Haifa RIMS marine workshop for their help in the field.

## Author Contributions

**Conceptualization:** Gilad Shtienberg.

**Data curation:** Gilad Shtienberg, Assaf Yasur-Landau, Richard D. Norris.

**Formal analysis:** Gilad Shtienberg.

**Funding acquisition:** Assaf Yasur-Landau, Thomas E. Levy.

**Investigation:** Gilad Shtienberg, Assaf Yasur-Landau, Michael Lazar.

**Methodology:** Gilad Shtienberg, Tammy M. Rittenour, Anthony Tamberino, Katrina Cantu, Ehud Arkin-Shalev, Steven N. Ward, Thomas E. Levy.

**Project administration:** Assaf Yasur-Landau.

**Software:** Gilad Shtienberg.

**Supervision:** Richard D. Norris, Tammy M. Rittenour, Thomas E. Levy.

**Validation:** Gilad Shtienberg, Richard D. Norris, Michael Lazar, Tammy M. Rittenour, Omri Gadol, Katrina Cantu.

**Visualization:** Gilad Shtienberg.

**Writing – original draft:** Gilad Shtienberg.

**Writing – review & editing:** Assaf Yasur-Landau, Richard D. Norris, Michael Lazar, Tammy M. Rittenour, Omri Gadol, Thomas E. Levy.

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
