## [Decision Letter · Decision Letter 0]

12 Oct 2020

PONE-D-20-29118

A Neolithic Mega-Tsunami Event in the Eastern Mediterranean: Prehistoric Settlement Resilience Along the Carmel Coast, Israel

PLOS ONE

Dear Dr. Shtienberg,

Thank you for submitting your manuscript to PLOS ONE. After careful consideration, we feel that it has merit but does not fully meet PLOS ONE’s publication criteria as it currently stands. Therefore, we invite you to submit a revised version of the manuscript that addresses the points raised during the review process.

We look forward to receiving your revised manuscript.

Kind regards,

Peter F. Biehl, PhD

Academic Editor

PLOS ONE

Journal Requirements:

2. In your Methods section, please provide additional information regarding the permits you obtained to collect samples for the present study.

Please ensure you have included the full name of the authority that approved the field site access and, if no permits were required, a brief statement explaining why.

'The Authors gratefully acknowledge the generously support provided by: Scripps Center for Marine Archaeology, Scripps Institution of Oceanography, UC San Diego; The Koret Foundation (Grant ID 19-0295); Murray Galinson San Diego – Israel Initiative; the Israel Institute (Washington, D.C.); Marian Scheuer-Sofaer and Abraham Sofaer Foundation; Norma and Reuben Kershaw Family Foundation; Ellen Lehman and Charles Kennel - Alan G Lehman and Jane A Lehman Foundation; Paul and Margaret Meyer and the Israel Science foundation (Grant ID 495/18).'

a. Please remove any funding-related text from the manuscript and let us know how you would like to update your Funding Statement. Currently, your Funding Statement reads as follows: 'No'

4. Thank you for stating the following in your Competing Interests section: 'No'

a. Please complete your Competing Interests statement to state any Competing Interests.

If you have no competing interests, please state "The authors have declared that no competing interests exist.", as detailed online in our guide for authors at http://journals.plos.org/plosone/s/submit-now

5. We note that Figures 1, 2 and 6 in your submission contain map/satellite images which may be copyrighted.

a. You may seek permission from the original copyright holder of Figures 1, 2 and 6 to publish the content specifically under the CC BY 4.0 license. 

Additional Editor Comments:

Your manuscript has now been seen by three referees, whose comments are appended below. You will see from these comments that while the referees find your work of great interest, they have raised concerns that must be addressed before re-submission.

Reviewers' comments:

Reviewer's Responses to Questions

**Comments to the Author**

1. Is the manuscript technically sound, and do the data support the conclusions?

Reviewer #1: Yes

Reviewer #2: Partly

Reviewer #3: Yes

2. Has the statistical analysis been performed appropriately and rigorously? 

Reviewer #1: Yes

Reviewer #2: Yes

Reviewer #3: N/A

3. Have the authors made all data underlying the findings in their manuscript fully available?

Reviewer #1: Yes

Reviewer #2: Yes

Reviewer #3: Yes

4. Is the manuscript presented in an intelligible fashion and written in standard English?

Reviewer #1: Yes

Reviewer #2: Yes

Reviewer #3: Yes

5. Review Comments to the Author

Reviewer #1: Manuscript Number: PONE-D-20-29118

Manuscript Title: A Neolithic Mega-Tsunami Event in the Eastern Mediterranean: Prehistoric Settlement Resilience Along the Carmel Coast, Israel

Review (28th Sep 2020)

This a very thoroughly designed piece of scientific research. It reports on a vast amount of field-work and which is presented in a compact and understandable manner. Congratulations ! In my function as reviewer I have some minor corrections and proposals, as follows:

(1) Time-units

(2) Titel and Conclusions

(3) Reference

(4) An Open Question

(1). Time Units:

In lines (inter alia): 224, 227, 242, 243, 245, 266,267,268

The use of time-scales ‘ya’ is non-standard and does not correspond to any known physical units of time measurements. Please replace by calBP. The same applies to the term ‘years ago’, for which the same applies. Both ‘ya’ and ‘years ago’ sound rather home-made.

(2). Titel and Conclusions:

Your use in the Titel of the buzzword ‘resilience’ contradicts your statement in the Conclusion that tsunamis played a ‘destabilizing role’. Either ‘resilience’ or ‘destabilizing’: Tertium non Datur. Looking closer, it becomes clear that you are differentiating between two periods, one (early Holocene) that has a lot of tsunamis, and the other (later Holocene) that has few tsunamis. A very nice result ! But, what this to do with ‘resilience’ or ’destabilization’ ? IF it is true – as you write – that the early Holocene PPNA-B settlements were mega-tsunami-destroyed, how can you know whether the corresponding societies were ‘tsunami-destabilized’ ? Similarly, IF it is true - as you write – that the many later PPNB-C sites indicate resettlement following the mega-tsunami, what can this have to do with ‘tsunami-resilience’ ? Does not the whole question boil down to what we know - or do not know - about societal reactions towards adverse conditions (whether fast or slow, whether climatic and/or society-internal). Of course, it is not your task to resolve these questions ! Even so, I think that the very word resilience is not particularly convincing , when applied to potential tsumami impact and societal consequences.

Proposal (in the Titel): Replace Resilience by Vulnerability.

(3). Reference

Line 386:

You write: Recovery from the early Holocene tsunami event was slow, possibly also affected by the 9,250 ya cold and arid climatic event [49].

But: reference [49] does not find evidence for impact of climate on prehistoric societies ! According to [49], prehistoric societies are resilient towards climate variability. If, at this point, you simply need a reference to demonstrate the existence of an 9,25 ka calBP climate event, and which is not cluttered up by the archaeological discussion of radiocarbon-based demography, you could take out [49] and replace it by Fleitmann & 2008:

https://agupubs.onlinelibrary.wiley.com/doi/abs/10.1029/2007PA001519

(4). Open Question

The following sentence is both misplaced and not understandable:

Line 376

By the PPNA and PPNB, the heartland of prehistoric settlement had shifted to the interior of the southern Levant [42,48].

True is (& perhaps you mean ?): the (well-known) major expansion both in size of sites as well as settlement area (call it ‘across the Jordan from West to East) at around 9.3 ka calBP, for which you might like to use as (earliest) reference the PhD by Hans-Georg Gebel (especially: Abb.3) :

https://freidok.uni-freiburg.de/data/466

Reviewer #2: I recommend that this article should be published to open up discussion on the topic. The core argument is tantalizing and partly speculative, despite rigorous OSL analysis, and it may be beneficial to hear counter-arguments based on other lines of evidence. For example, does the proposed paleotsunami evidence indicate such a devastating impact that the entirety of ca. 4000 years of settlement evidence would have been completely destroyed? Perhaps one might expect even ephemeral traces of remaining architecture (slight though such architecture may have been in the LN/PPNA), or some surviving skeletal evidence (e.g. complete animal and/or human skeletons which might indicate the sudden impact of the tsunami; or injury evidence, and so on). Evidence for the lack of Late/Final Natufian and PPNA sites (which is not uncommon throughout the southern Levant at this time) could also be due to landscape/settlement practices by human communities, or to archaeological survey biases. All these factors would have to be discounted before a firm conclusion based on such a very localized event could be substantiated. But overall, this is a fascinating piece of research which deserves to be fully debated.

Reviewer #3: This paper presents evidence for a tsunami event at Dor in the Eastern Mediterranean, dated by OSL to have occurred at some point between 9910 and 9290 BP, or during the Pre-Pottery Neolithic B. This is very interesting information and this type of evidence is often overlooked in archaeology especially. The authors have studied the geological (and more, e.g. faunal) evidence extensively, and present detailed data. In total 8 OSL dates were done; the event is dated by 3 of those, so it would be good to confirm the dating further in future and especially to get a narrower range, but for now the dates at least are convincing and considering how difficult it is to date these events it is already adding much to our knowledge just with these dates. I therefore recommend publishing this paper, although I have a few comments below that I would like to see addressed first.

Comments:

1) My main comment is on the interpretation of how the tsunami event might have affected the communities living in the area. In terms of direct impacts on these communities and on the preservation of earlier sites, I agree that the sites in the area covered by the water would have been destroyed and eroded so that now they are not found, and this is a very interesting conclusion in itself. However, I am less convinced by taking the argument further to ascribe larger-scale socio-economic and population growth implications to the tsunami event. While it is certainly of interest to bring these ideas up, I think they should be discussed more carefully, taking into account: 1) the large range of the dating, so that it is difficult to be sure about the synchronicity between the tsunami and described changes in settlement patterns (e.g. the tsunami might have occurred only after the inland MPPNB sites grew; or it might have occurred only shortly before the LPPNB so that the impact was not long term, contra line 386); and 2) the area actually reconstructed to be destroyed, 1.5-3.5km inland in one area of the coast, relatively to the rest of the ‘Neolithic’ area. As such I would argue it goes too far to claim that the event would have caused “a reversal of the Neolithic Demographic Transition” (line 373) or to imply that absence of tsunami events allowed for “unhindered growth” afterwards (line 417-419).

2) Related to the above, I would also like to see more on where the site pattern information is based on. For example in Figure 2 it is very useful that submerged PPNC and PN sites are indicated, and in addition it would be very useful to see also the other, inland known sites by period. Currently five such sites are indicated but not coded by period, and it would be good to know if, and where, other sites of the period are within the map area (or, for which area on the map the ‘complete’ settlement pattern is given). Perhaps this could be a separate figure if this one gets too busy otherwise. In addition, it would be good to briefly discuss the background of the settlement pattern, and mainly if it could be affected by a research bias (e.g. a large research project focusing on a specific sub-area, or extensive building work and therefore rescue archaeology in certain places) or not. It is no problem if this turns out to be the case, it just needs to be taken into account.

3) Consistency in dates: Throughout the paper the way dates are presented should be made consistent. Currently yr, ka yr, years ago, calibrated years before present, and BC/AD (Supplementary Tables 1+2) are all used, which makes it confusing to the reader. The dating used for the archaeological periods in the Supplementary Information should also be the same as in the main text and throughout the text.

Specific, minor comments:

Line 58-60. Not so much rare in general, but for tsunami events I agree, although it might be worth citing some studies that have looked at this (e.g. Waddington and Wicks 2017 Journal of Archaeological Science).

Figure 1 and caption: it would be useful to have the names of the numbered sites here in the figure or in the caption, not only in the Supplementary Information.

Line 107-109: I would probably move this sentence to the Materials and Methods section and instead add here a sentence specifically stating the aims/objectives of the study.

Material and Methods:

-OSL dating: Was an age model done? If not, why not? And if yes, how was it done?

-Table 1: Please present the dates in a consistent way. It says in the caption they are in ka years, but in the table they are partly in years and partly in ka year.

Table 2: This could perhaps go in the Supplementary Information.

Lines 219-221: repetition, already stated in the Methods section

Line 424: “earliest” I suggest “earliest known”, there are probably earlier, yet undocumented ones

Supplementary Information:

-Very useful in general.

-What is the difference between Table S1 and S2? Why are these not in one table?

Because PLOS ONE does not do copy-editing, I add a few notes on typos and spelling here:

-Remove hyphens between adjectives and nouns, e.g. eastern-Mediterranean should be eastern (or Eastern) Mediterranean, early-Holocene should be early Holocene, marine-sand should be marine sand, and so on.

-Line 77: “Less” should be with lower case l

-“Suplement matirial” should be “Supplemental material” (or as called in PLOS ONE “Supporting Information”).

6. PLOS authors have the option to publish the peer review history of their article (what does this mean?). If published, this will include your full peer review and any attached files.

Reviewer #1: No

Reviewer #2: No

Reviewer #3: No

---

## [Author Response · Author response to Decision Letter 0]

4 Nov 2020

Reviewer 1 comments

1. The use of time-scales 'ya' is non-standard and does not correspond to any known physical units of time measurements. Please replace by calBP. The same applies to the term 'years ago', for which the same applies. Both 'ya' and 'years ago' sound rather home-made.

The use of calBP is reserved for radiocarbon dating, and since our radiometric dating technique relies solely on optically stimulated luminance, we used ka (kilo-annum) (Nelson et al., 2015) throughout the revised manuscript and figures.

2. Your use in the Titel of the buzzword 'resilience' contradicts your statement in the Conclusion that tsunamis played a 'destabilizing role'. Either 'resilience' or 'destabilizing': Tertium non Datur. Looking closer, it becomes clear that you are differentiating between two periods, one (early Holocene) that has a lot of tsunamis, and the other (later Holocene) that has few tsunamis. A very nice result ! But, what this to do with 'resilience' or' destabilization'? IF it is true – as you write – that the early Holocene PPNA-B settlements were mega-tsunami-destroyed, how can you know whether the corresponding societies were 'tsunami-destabilized'? Similarly, IF it is true - as you write – that the many later PPNB-C sites indicate resettlement following the mega-tsunami, what can this have to do with 'tsunami-resilience'? Does not the whole question boil down to what we know - or do not know - about societal reactions towards adverse conditions (whether fast or slow, whether climatic and/or society-internal). Of course, it is not your task to resolve these questions ! Even so, I think that the very word resilience is not particularly convincing , when applied to potential tsumami impact and societal consequences.

After carefully examining this comment, we agree with Reviewer 1 and replaced the word Resilience with Vulnerability in the title.

3. Reference Line 386: you write: Recovery from the early Holocene tsunami event was slow, possibly also affected by the 9,250 ya cold and arid climatic event [49]. But: reference [49] does not find evidence for impact of climate on prehistoric societies ! According to [49], prehistoric societies are resilient towards climate variability.

We agree with the statement of Reviewer 1 and erased the sentence, "possibly also affected by the 9.25 ka cold and arid climatic event [49]" from the manuscript.

4. The following sentence is both misplaced and not understandable: Line 376: By the PPNA and PPNB, the heartland of prehistoric settlement had shifted to the interior of the southern Levant [42,48]. True is (& perhaps you mean ?): the (well-known) major expansion both in size of sites as well as settlement area (call it 'across the Jordan from West to East) at around 9.3 ka calBP,

According to the comment raised by Reviewer 1, we changed the sentences as follows: 

The rapid growth in size and area of autonomous Middle PPNB settlements in the hill country of Israel and across the Jordan area may reflect the move away from the coastal lowlands [47,48]. 

Reviewer 2 comments

1. Does the proposed paleotsunami evidence indicate such a devastating impact that the entirety of ca. 4000 years of settlement evidence would have been completely destroyed? Perhaps one might expect even ephemeral traces of remaining architecture (slight though such architecture may have been in the LN/PPNA), or some surviving skeletal evidence (e.g. complete animal and/or human skeletons which might indicate the sudden impact of the tsunami; or injury evidence, and so on). Evidence for the lack of Late/Final Natufian and PPNA sites (which is not uncommon throughout the southern Levant at this time) could also be due to landscape/settlement practices by human communities, or to archaeological survey biases. All these factors would have to be discounted before a firm conclusion based on such a very localized event could be substantiated.

The issue of the identification of paleo-tsunami deposits in the geological record, as well as its possible impact on the landscape and ancient societies, is a highly charged issue in the earth and social sciences. Accordingly, we began our research on this issue with great skepticism. From the time we extracted the cores in the field in Israel in July 2018 to the analyses of the cores here at UCSD – SIO, we have been very cautious in our interpretation, only finalizing the results after obtaining the radiometric dates of the poorly sorted quartz sand layer that consists of marine mollusk shells limestone pebbles, calcareous sandstone clasts, and dark, silty clay rip-up clasts as well as the overlying and underlying units. Once we were confident with our interpretation, we used the analyses of the depositional patterns in the Dor sediment cores to provide insights for identifying how the ancient tsunami impacted the prehistoric settlement patterns of the Carmel coast.

We raise our hypothesis regarding the impact of the paleo-tsunami on coastal societies after carefully considering the possible outcomes of such a devastating event on the low-lying areas located a few kilometers from the present shoreline. The consequence of such a tsunami may have dramatically reduced the carrying capacity of the Carmel coast's lowland areas by increasing the salt concentrations of the soils and underlying aquifer and destroying fields, pasture, and herds leading to significant economic disruption. 

The strong catastrophic event is also believed, similar to modern-day incidents, to have had a profound impact on the landscape, eroding a few tens of centimeters from the paleo-surface (Chagué-Goff et al., 2011; Paris et al., 2009; Yamada et al., 2014). This known phenomenon could explain why numerous expeditions that have taken place over the last decades in the shallow marine and coastal parts of the Carmel coast (Barkai and Biran, 2012; Edwards, 2016; Galili et al., 2019; Galili et al., 1993; Galili and Nir, 1993; Galili et al., 1997; Galili and Weinstein-Evron, 1985; Goring-Morris, 1984; Meier et al., 2017) have not found Natufian – Pre pottery Neolithic B sites nor human remains from the lowlands while the existence of such sites has been identified in the Carmel coast in areas with elevations that are higher than 20 m with respect to present-day mean sea level. We hope this addresses the query of Reviewer 2.

Reviewer 3 comments

1. The large range of the dating, so that it is difficult to be sure about the synchronicity between the tsunami and described changes in settlement patterns (e.g. the tsunami might have occurred only after the inland MPPNB sites grew; or it might have occurred only shortly before the LPPNB so that the impact was not long term.

Please refer to our response that answered comment 1 raised by Reviewer 2.

2. The area actually reconstructed to be destroyed, 1.5-3.5km inland in one area of the coast, relatively to the rest of the 'Neolithic' area. As such I would argue it goes too far to claim that the event would have caused "a reversal of the Neolithic Demographic Transition" (line 373) or to imply that absence of tsunami events allowed for "unhindered growth" afterwards.

We agree with this comment and changed the text accordingly:

"With carrying capacity and fertility decreased, population mobility is thought to increase, influencing the Neolithic demographic transition and possibly decreasing population density".

.

3. Related to the above, I would also like to see more on where the site pattern information is based on. For example in Figure 2 it is very useful that submerged PPNC and PN sites are indicated, and in addition it would be very useful to see also the other, inland known sites by period. Currently five such sites are indicated but not coded by period, and it would be good to know if, and where, other sites of the period are within the map area (or, for which area on the map the 'complete' settlement pattern is given). Perhaps this could be a separate figure if this one gets too busy otherwise. In addition, it would be good to briefly discuss the background of the settlement pattern, and mainly if it could be affected by a research bias (e.g. a large research project focusing on a specific sub-area, or extensive building work and therefore rescue archaeology in certain places) or not. It is no problem if this turns out to be the case, it just needs to be taken into account.

Considering comment 3 we have color-coded the known terrestrial Natufian – PPNC sites in figure 2 and made the necessary changes in the figure caption and text.

4. Consistency in dates: Throughout the paper the way dates are presented should be made consistent. Currently yr, ka yr, years ago, calibrated years before present, and BC/AD (Supplementary Tables 1+2) are all used, which makes it confusing to the reader. The dating used for the archaeological periods in the Supplementary Information should also be the same as in the main text and throughout the text.

In accordance with comment four, we have changed the temporal unit for the OSL ages to ka (kilo-annum) in the text, tables, and figures. Additionally, we have added the temporal unit ka to the 14C calibrated dates to text and figures.

5. Not so much rare in general, but for tsunami events I agree, although it might be worth citing some studies that have looked at this (e.g. Waddington and Wicks 2017 Journal of Archaeological Science).

The word "rare" was changed into "less common," and the reference was added to the manuscript.

6. Figure 1 and caption: it would be useful to have the names of the numbered sites here in the figure or in the caption, not only in the Supplementary Information.

In accordance with comment 6 we added the site names to the figure caption.

7. Line 107-109: I would probably move this sentence to the Materials and Methods section and instead add here a sentence specifically stating the aims/objectives of the study.

The last sentence found in the introduction was removed from the text, and as Reviewer 3 suggested, the main objective of this study was added:

"The preservation of Quaternary deposits and landforms at Dor coupled with the abundance archaeological sites provide a unique opportunity to investigate the geomorphological processes and human-landscape interactions in the eastern Mediterranean".

8. OSL dating: Was an age model done? If not, why not? And if yes, how was it done?

Equivalent Dose (DE) was calculated for each OSL age using the central age model (CAM) for all samples. Aliquots were rejected if they had evidence of feldspar contamination, recycling ratio <0.1 or >1.1, recuperation >10% of the natural signal, or natural DE greater than the highest regenerative dose given. Errors on DE values are reported at 2-sigma standard error, and age estimates are reported at 1-sigma standard error (see Table 1). Uncertainties include errors related to instrument calibration, dose rate, and equivalent dose calculations and calculated in quadrature (Aitken, 2003; Guérin et al., 2011). This explanation is presented in the extended methods section found in the supporting information file.

9. Table 1: Please present the dates in a consistent way. It says in the caption they are in ka years, but in the table they are partly in years and partly in ka year.

The units for time measurements presented in the manuscript figure and tables are all presented in a similar - ka (kilo-annum).

10. Table 2: This could perhaps go in the Supplementary Information.

The research that is presented in our manuscript relies on the OSL ages that we have generated in our study. Because these ages are such an essential part of the investigation, we think that both Tables 1 and 2, which present the optically stimulated measurements, are needed in the main text and will benefit the readers.

11. Lines 219-221: repetition, already stated in the Methods section

This line was erased from the manuscript.

12. Line 424: "earliest" I suggest "earliest known", there are probably earlier, yet undocumented ones

The word "known" was added to the sentence.

13. What is the difference between Table S1 and S2? Why are these not in one table?

These two tables present a compilation of prehistoric - historic tsunami deposits (Table S1) and documented Tsunami events (Table S2) from the eastern Mediterranean. And thus, we think that should be kept separate.

14. Remove hyphens between adjectives and nouns, e.g. eastern-Mediterranean should be eastern (or Eastern) Mediterranean, early-Holocene should be early Holocene, marine-sand should be marine sand.

These grammar mistakes were amended.

15. Line 77: "Less" should be with lower case

This spelling error was corrected.

16. "Suplement matirial" should be "Supplemental material" (or as called in PLOS ONE "Supporting Information").

The name of the file was changed accordingly.

References

Aitken, M., 2003. Radiocarbon dating. Archaeological Method and Theory, 505-508.

Barkai, R., Biran, N., 2012. Aviel: a new Neolithic site at the foothills of Mt. Carmel. NEO-LITHICS 2/11 14.

Chagué-Goff, C., Schneider, J.-L., Goff, J.R., Dominey-Howes, D., Strotz, L., 2011. Expanding the proxy toolkit to help identify past events — Lessons from the 2004 Indian Ocean Tsunami and the 2009 South Pacific Tsunami. Earth-Science Reviews 107, 107-122.

Edwards, P.C., 2016. The chronology and dispersal of the Pre-Pottery Neolithic B cultural complex in the Levant. Paléorient, 53-72 %@ 0153-9345.

Galili, E., Benjamin, J., Eshed, V., Rosen, B., McCarthy, J., Kolska Horwitz, L., 2019. A submerged 7000-year-old village and seawall demonstrate earliest known coastal defence against sea-level rise. PLoS One 14, e0222560.

Galili, E., Dahari, U., Sharvit, J., 1993. Underwater surveys and rescue excavations along the Israeli coast. The International Journal of Nautical Archaeology 22, 61-77.

Galili, E., Nir, Y., 1993. The submerged Pre-Pottery Neolithic water well ofAtlit-Yam, northern Israel and its palaeoenviromental implications. The Holocene 3, 265-270.

Galili, E., Stanley, D.J., Sharvit, Y., Weinstein‐Evron, M., 1997. Evidence for Earliest Olive-Oil Production in Submerged settlments off the Carmel Coast, Israel. Tournal of archaeologica Science 24, 1141-1150.

Galili, E., Weinstein-Evron, M., 1985. Prehistory and paleoenvironments of submerged sites along the Carmel coast of Israel. Paleorient 11, 37-52.

Goring-Morris, A.N., 1984. The lithic assemblages from Tel Mevorakh. Excavations at Tel Mevorakh (1973-1976), Qedem 18, 81-86.

Guérin, G., Mercier, N., Adamiec, G., 2011. Dose-rate conversion factors: update. Ancient TL 29, 5-8.

Meier, J.S., Goring-Morris, A.N., Munro, N.D., 2017. Aurochs bone deposits at Kfar HaHoresh and the southern Levant across the agricultural transition. antiquity 91, 1469-1483 %@ 0003-1598X.

Nelson, M.S., Gray, H.J., Johnson, J.A., Rittenour, T.M., Feathers, J.K., Mahan, S.A., 2015. User Guide for Luminescence Sampling in Archaeological and Geological Contexts. Advances in Archaeological Practice 3, 166-177.

Paris, R., Wassmer, P., Sartohadi, J., Lavigne, F., Barthomeuf, B., Desgages, E., Grancher, D., Baumert, P., Vautier, F., Brunstein, D., Gomez, C., 2009. Tsunamis as geomorphic crises: Lessons from the December 26, 2004 tsunami in Lhok Nga, West Banda Aceh (Sumatra, Indonesia). Geomorphology 104, 59-72.

Yamada, M., Fujino, S., Goto, K., 2014. Deposition of sediments of diverse sizes by the 2011 Tohoku-oki tsunami at Miyako City, Japan. Marine Geology 358, 67-78.

---

## [Decision Letter · Decision Letter 1]

25 Nov 2020

A Neolithic Mega-Tsunami Event in the Eastern Mediterranean: Prehistoric Settlement Vulnerability Along the Carmel Coast, Israel

PONE-D-20-29118R1

Dear Dr. Shtienberg,

We’re pleased to inform you that your manuscript has been judged scientifically suitable for publication and will be formally accepted for publication once it meets all outstanding technical requirements.

Kind regards,

Peter F. Biehl, PhD

Academic Editor

PLOS ONE

Additional Editor Comments (optional):

Reviewers' comments:

Reviewer's Responses to Questions

**Comments to the Author**

1. If the authors have adequately addressed your comments raised in a previous round of review and you feel that this manuscript is now acceptable for publication, you may indicate that here to bypass the “Comments to the Author” section, enter your conflict of interest statement in the “Confidential to Editor” section, and submit your "Accept" recommendation.

Reviewer #1: All comments have been addressed

Reviewer #2: All comments have been addressed

Reviewer #3: All comments have been addressed

2. Is the manuscript technically sound, and do the data support the conclusions?

Reviewer #1: Yes

Reviewer #2: (No Response)

Reviewer #3: Yes

3. Has the statistical analysis been performed appropriately and rigorously? 

Reviewer #1: Yes

Reviewer #2: (No Response)

Reviewer #3: Yes

4. Have the authors made all data underlying the findings in their manuscript fully available?

Reviewer #1: Yes

Reviewer #2: (No Response)

Reviewer #3: Yes

5. Is the manuscript presented in an intelligible fashion and written in standard English?

Reviewer #1: Yes

Reviewer #2: (No Response)

Reviewer #3: Yes

6. Review Comments to the Author

Reviewer #1: (No Response)

Reviewer #2: (No Response)

Reviewer #3: (No Response)

7. PLOS authors have the option to publish the peer review history of their article (what does this mean?). If published, this will include your full peer review and any attached files.

Reviewer #1: No

Reviewer #2: No

Reviewer #3: No

---

## [Editor Report · Acceptance letter]

1 Dec 2020

PONE-D-20-29118R1 

A Neolithic Mega-Tsunami Event in the Eastern Mediterranean: Prehistoric Settlement Vulnerability Along the Carmel Coast, Israel. 

Dear Dr. Shtienberg:

I'm pleased to inform you that your manuscript has been deemed suitable for publication in PLOS ONE. Congratulations! Your manuscript is now with our production department. 

Kind regards, 

on behalf of

Dr. Peter F. Biehl 

Academic Editor

PLOS ONE